# Arterial Resections in Pancreatic Cancer—An Updated Systematic Review and Meta-Analysis

**DOI:** 10.3390/cancers17091540

**Published:** 2025-05-01

**Authors:** Colin Noel, Adeboye Azeez, Annamarie Du Preez, Kiera Noel

**Affiliations:** 1Division of Gastrointestinal Surgery, Department of Surgery, Faculty of Health Sciences, University of the Free State, Bloemfontein 9301, South Africa; 2Gastrointestinal Research Unit, University of the Free State, Bloemfontein 9301, South Africa; azeez.an@ufs.ac.za; 3Faculty of Health Sciences, University of the Free State, Bloemfontein 9301, South Africa; annamari@ufs.ac.za (A.D.P.); noelkmm@ufs.ac.za (K.N.)

**Keywords:** arterial resection, vascular resection, pancreatic cancer, pancreatic neoplasm, mortality, pancreaticoduodenectomy, pancreatectomy

## Abstract

Pancreatic cancer has one of the worst prognoses of all malignancies. Despite complete surgical resection of pancreatic cancer being the mainstay of curative treatment, only a minority of patients qualify for surgery. Extending surgery to include resection of involved arteries in pancreatic cancer surgery will increase the number of patients who can qualify for surgery but data on the benefit and safety of arterial resections are still under investigation. Previous studies showed a decrease in the morbidity and mortality associated with arterial resections in pancreatic cancer over time and suggested that arterial resections may become a feasible treatment option in the future. This updated systematic review and meta-analysis on arterial resections in pancreatic cancer aims to understand the morbidity and mortality associated with arterial resections in the modern era incorporating recent studies published from 2018 to 2024.

## 1. Introduction

Pancreatic cancer mortality has increased in the United States over the past decade and is the third leading cause of cancer-related mortality in men and women combined [1]. Despite modern chemotherapy and surgery, it continues to have one of the worst 5-year survival rates of ~10% [1].

Long-term survival can only be achieved through a combination of complete oncological surgical R0 resection and chemotherapy. The proximity of major vessels together with the local invasive nature of pancreatic cancer has, in the past, limited the ability to achieve complete surgical excision in many cases. Up to 25% of patients with pancreatic cancer will have vascular involvement. Historically, associated vascular involvement was a contraindication of surgical resection in pancreatic cancer [2]. Current guidelines support venous resections [3], with the recent literature showing comparative overall survival rates to those undergoing standard resections without venous resections [4]. In addition, vascular resections are associated with an increased R0 resection rate which continues to be one of the most important factors contributing to a favorable prognosis, even in borderline resectable and locally advanced pancreatic cancer [5]. The notion of arterial resections in the setting of pancreatic cancer, however, is more controversial. A recent systematic review and meta-analysis on arterial resections in pancreatic cancer [6] using studies published from 1991 to 2018 showed that arterial resections were associated with a greater risk of death (RR: 4.09; *p* < 0.001) and complications (RR: 1.4; *p* = 0.01). Subgroup analysis on complications, however, showed that the overall increased morbidity was driven by post-operative bleeding with no differences in post-operative biliary or pancreatic fistulas, and cardiovascular or pulmonary complications. The authors concluded by stating that arterial resections in pancreatic cancer may nevertheless become a viable treatment option. The recommendation despite a worse overall survival when compared to venous resections may, in part, be due to the limited alternative options and prognosis associated with palliative treatment only.

With the continued expansion of surgical technique and arterial resections in an attempt to improve survival outcomes, the aim of this meta-analysis was to assess the modern outcomes of arterial resections for pancreatic cancer since the publication of Małczak et al. [6] from 2018 to 2024.

## 2. Materials and Methods

This meta-analysis was conducted following PRISMA guidelines (Appendix A), and the study protocol was registered in the PROSPERO database (CRD420251021979).

### 2.1. Search Strategy

This systematic review follows the standard methods outlined in the Preferred reporting items for systematic review and meta-analysis of diagnostic test accuracy studies (PRISMA-DTA) guidelines as specified in the explanation, elaboration, and checklist [7]. A comprehensive search strategy was developed to identify all relevant articles published between January 2018 and November 2024 to examine the differences between arterial and non-arterial pancreatic resections. The databases searched included MEDLINE, SCOPUS, and Web of Science Core Collection (WoS). The search strategy was developed using a combination of relevant keywords and filters. Key search terms included “pancreatic cancer*” OR “Pancreatic Neoplasm*” OR “pancreatic adenocarcinoma*” AND pancreaticoduodenectomy OR Whipple OR pancreatectomy OR “pancreatic surger*” OR “triangle operation*” OR “triangle surger*” AND morbidity OR mortality. The search results were managed using Rayyan web and mobile app for systematic reviews [8]. After removing duplicates, the full texts of all potentially relevant articles were retrieved and reviewed to ensure they met the inclusion criteria.

### 2.2. Eligibility Assessment

Records identified through the search strategy were thoroughly assessed by the review team. Each abstract and full-text article was independently evaluated by three reviewers (AA, KN, and CN). In cases of disagreement, consensus was achieved through discussion among the entire review team. Only studies that included a comparison of mortality or morbidity were included. We excluded animal studies, conference abstracts, reviews, video articles, case reports, studies lacking relevant data (i.e., those not comparing arterial versus non-arterial pancreatic resections), and studies not published in English.

### 2.3. Study Selection

Relevant data were independently extracted by two reviewers (AA and KN). In cases of disagreement in the extracted data, a third reviewer (CN) was consulted to reach consensus. Disagreements in the selection process were resolved through discussion among all the reviewers (AA, AD, KN, and CN). Only full-text articles were included in the extraction process. The following information was extracted from each study: first author, year of publication, study design, number of patients, age, sex, surgery type, type of pancreatic resection, morbidity (presence of post-operative complications), mortality (30-days post-operative death), length of hospital stay (LOS), Clavien–Dindo classification (CDC ≥ 3), 1-, 2-, and 3-year survival rates, neoadjuvant treatment (NAT), R0 resection rate, and oncological outcomes of the surgery. A flowchart of the study selection process is presented in Figure 1.

### 2.4. Quality Assessment

We assessed the quality of the included non-randomized studies using the Newcastle–Ottawa Scale (NOS) for Cohort Studies [9]. The NOS evaluates three key domains: patient selection, comparability of study groups, and outcome assessment. Each study was rated on a scale of 0 to 9, with a score of 6 or higher considered to indicate high quality.

### 2.5. Statistical Analysis

Statistical analyses were performed using the R programming language (version 4.4.1) in RStudio 2024.12.0 + 467 for macOS after data were imported from Microsoft Excel 356. The prevalence of mortality, morbidity, and overall survival for arterial and non-arterial resections was evaluated using a random effects model to calculate pooled prevalence and the corresponding 95% confidence interval (CI), which were subsequently illustrated in a forest plot. The degree of heterogeneity among the studies was assessed using the I^2^ statistic. The I^2^ values were interpreted using specific thresholds: 0–25%, 25–50%, 50–75%, with values greater than 75% indicating inconsequential, low, moderate, and high heterogeneity [10]. To ensure consistency, a sensitivity analysis was performed by systematically excluding individual studies. Publication bias was assessed using funnel plots. A *p*-value < 0.05 was considered indicative of statistical significance.

## 3. Results

Initially, a total of 1903 articles were identified and retrieved from various databases. After removing 226 duplicates, an additional 968 records were excluded due to being review articles, case reports, conference abstracts, non-English publications, animal studies, or lacking relevant outcome comparisons between arterial and non-arterial resections. A total of 709 articles were screened based on titles and abstracts, resulting in 22 papers selected for eligibility with full-text review. Following full-text evaluation, seven studies were deemed eligible for data extraction. Our review includes 5465 patients, with 206 undergoing arterial resections and 4192 undergoing pancreatic resections without arterial involvement. The characteristics of the included studies are outlined in Table 1. These articles were published between 2018 and 2024. There were two articles each from China and Germany, and one article each from Denmark, Japan, and USA.

Among the seven studies included in this review, six studies reported morbidity [11,12,13,14,15,16], four studies reported mortality [12,13,15,16], and four studies reported overall survival [11,13,15]. Six studies used a retrospective study design, while only one study used a prospective study design. Overall, the quality of the included studies is moderate, with the majority scoring at least 6 points on the Newcastle–Ottawa Scale (NOS) (Appendix A).

**Table 1 cancers-17-01540-t001:** Baseline characteristics of studies included in the systematic review and meta-analysis.

First Author	Year	Study Type	Total	Country	Sex: M/F	Mean Age	Types of Operation (Surgery)	Type of Surgical Approach	Complications	NOS
Loos [16]	2024	R	2135	Germany	659/793	61.5/62.3	DP (2135, 100%)	Open/LAP/ROB	POPF/DGE/NAT/PPH/CDC/LOS	7
Asano [17]	2018	R	51	Japan	27/24	73/73	PD (51, 100%)	NR	BL/CDC/LOS	4
Yang [11]	2019	R	56	China	32/24	63.5/63.8	PD (44, 78.5%), TP (12, 22.2%)	Open/LAP/ROB	POPF/DGE/BL/NAT/PPH/CDC/LOS/R0	8
Li [15]	2021	R	92	China	54/38	63.1/61.5	DP (92, 100%)	NR	POPF/DGE/BL/NAT/PPH/CDC/LOS	9
Malinka [13]	2020	R	40	Germany	22/18	65.05/60.75	DP (40, 100%)	Open/LAP/ROB	POPF/DGE/NAT/PPH/CDC/LOS	8
Storkholm [14]	2020	P	89	Denmark	50/39	62.3/58.3	DP (89, 100%)	Open/LAP/ROB	POPF/DGE/BL/NAT/PPH/CDC/LOS/R0	6
Zettervall [12]	2019	R	3002	USA	1420/1198	64/63	PD (3002, 100%)	Open/LAP/ROB	POPF/DGE	6

R, Retrospective; P, Prospective; A, Arterial; NA, Non-arterial; MOT, Mean operative time; DP, Distal pancreatectomy; PD, pancreaticoduodenectomy; LAP, laparoscopic approach; ROB, robotic; POPF, post-operative pancreatic fistula; DGE, delayed gastric emptying; NAT, neoadjuvant chemotherapy treatment; PPH, post-pancreatectomy haemorrhage; CDC, Clavien–Dindo classification; LOS length of hospital stay; R0, Negative margin resection rate; NOS, Newcastle–Ottawa Scale.

### 3.1. Morbidity

Morbidity was reported in six studies [11,12,13,14,15,16]. Zettervall and Loos contributed the most weight to the pooled estimate in the random effects model. The pooled analysis found no significant increase in morbidity rates between arterial resections (74/271, 27.31%) compared to non-arterial resections (665/4815, 13.81%). The risk ratios indicated a higher risk of morbidity in arterial resection compared to standard resection (RR: 1.48; 95% CI [1.16–1.89]; *p* = 0.2923), though this difference was not statistically significant. The studies showed low heterogeneity (I^2^ = 18.6%), suggesting minimal variation in the results across the studies (Figure 2).

### 3.2. Mortality

Mortality was reported in four studies [12,13,15,16]. Zettervall and Loos had the greatest influence on the pooled estimate in the random effects model. The arterial resections were significantly associated with a higher risk of mortality (11/173, 6.4%) compared to standard resections (75/4115, 1.8%). The risk ratios associated with arterial resection were 3-fold higher than standard resections (RR: 3.28; 95% CI [0.75–14.46]; *p* = 0.0365). There was, however, a substantial heterogeneity (I^2^ = 64.8%) shown across the studies (Figure 2).

### 3.3. Survival

The 1-, 2-, and 3-year survival rates for both arterial and standard resections were reported by three authors [11,13,15] (Table 2). Only (Li et al.) reported median overall survival for concomitant arterial and non-arterial resection at 27.4 months and 32.6 months, respectively. This suggests that concomitant arterial resections may result in slightly shorter or similar median survival durations, but the lack of median survival data reported in studies makes it difficult to draw definitive conclusions. For 1-year survival, arterial resections had survival rates ranging from 81.8% (Yang et al.) to 88% (Malinka et al.). Non-arterial standard resections from the same studies ranged from 74% (Yang et al.) to 94.3% (Li et al.). This indicates that arterial resections carry a similar risk of early mortality compared to non-arterial resections (Table 2). The overall 2-year survival rates after concomitant arterial resection ranged from 45% (Li et al.) to 78% (Malinka et al.) with a mean of 62.2%, while non-arterial resections ranged from 40% (Yang et al.) to 67.1% (Li et al.) with a mean of 55.7%. This suggests that arterial resections may lead to more consistent long-term survival benefits, although the mean difference between the two groups was not significantly different [mean difference: 6.5, 95%CI: (−55.42–68.42)] (Table 3). The overall 3-year survival rates after concomitant arterial resection ranged from 42.4% (Yang et al.) to 78% (Malinka et al.) with a mean of 50.1% from the three included studies. Non-arterial standard resections’ 3-year survival rates ranged from 38% (Li et al.) to 60% (Malinka et al.) with a mean of 49.8% (Table 2). For 3-year survival, there was no significant difference between the groups [mean difference: 0.333, 95%CI: (−49.38–50.04)] (Table 3).

### 3.4. Blood Loss

Blood loss was reported in four studies [11,14,15,17]. Asano had the greatest influence on the weight estimate in the effects model. The pooled data showed no significant differences between the arterial and standard resection (Mean Difference 32.683 mls; 95% CI: [−285.89–220.53]; *p* = 0.709); however, the heterogeneity was very high (I^2^ = 84.8%) across the studies (Table 3). The mean blood loss for arterial resections across the four studies was (733.4 mL) and standard resection (700.7 mL). The analysis showed no significant difference in blood loss between the two types of resections. The studies showed a high heterogeneity (I^2^ = 91.5%) across the studies (Appendix A).

### 3.5. Clavien–Dindo Classification (CDC) ≥ 3

CDC ≥ 3 was reported in six studies [11,13,17]. There were statistically no significant differences between the standard and arterial resection groups in terms of major complications (Mean Difference 2.6; 95% CI: [−21.52–16.32]; *p* = 0.738) (Table 3). However, the heterogeneity was very high (I^2^ = 94.9%) across studies (Appendix A).

### 3.6. Hospital Length of Stay (LOS)

LOS was reported in six studies [11,13,17]. The pooled data showed no significant differences in length of hospital stay between the arterial and standard resection (MD 0.383 days; 95% CI: [−0.736–0.866]; *p* = 0.874) (Table 3). The results suggest that the type of resection does not have a significant impact on hospital stay duration. The heterogeneity was, however, very high (I^2^ = 84.8%) across studies. The sensitivity analysis performed indicated that two studies were responsible for the high heterogeneity, Asano et al. [17] and Malinka et al. [13]. Excluding these studies has no effect on the results (Appendix A).

### 3.7. Neoadjuvant Chemotherapy Treatment (NAT)

NAT was reported in five studies [11,13,16]. Arterial resections were more significantly associated with administration of NAT (68/213, 31.9%) compared to standard resections (123/1707, 7.2%). This association was 2-fold higher and reflects the clinical practice of offering NAT to patients with arterial resection rather than standard resections (RR: 2.34; 95% CI [0.76–7.17]; *p* < 0.0001). The studies showed high heterogeneity (I^2^ = 84.2%) (Figure 3).

### 3.8. Delayed Emptying (DGE)

DGE was reported in six studies [11,12,13,14,15,16]. Arterial resections were associated with a significantly higher risk of DGE (29/226, 12.8%) compared to standard resections (465/4615, 10.1%). The risk ratios associated with arterial resection were 2-fold higher than standard resections (RR: 2.00; 95% CI [1.02–3.94]; *p* < 0.0054). The studies showed substantial heterogeneity (I^2^ = 69.9%) (Figure 3).

### 3.9. Post-Pancreatectomy Hemorrhage (PPH)

There were statistically no significant differences associated with arterial resections in terms of PPH (RR: 1.92; 95% CI [0.89–4.15]; *p* = 0.2471). There was low heterogeneity (I^2^ = 26.2%) across the studies (Figure 3).

### 3.10. Post-Operative Pancreatic Fistula (POPF)

POPF was reported in six studies [11,12,13,14,15,16]. Zettervall and Loos contributed the most weight to the pooled estimate in the random effects model. There were no statistically significant differences associated with arterial resections (RR: 1.20; 95% CI [0.93–1.56]; *p* = 0.974). There was also no heterogeneity (I^2^ = 0.0%) across the studies (Figure 3).

### 3.11. R0 Resection

R0 was reported in four studies [11,13,14,17]. The mean R0 resection rate in the four included studies in the arterial group was 68.9% and 76.6% in the standard resection group. Yang and Storkholm contributed the most weight to the pooled estimate in the random effects model. The pooled analysis indicated the risk ratio of achieving a R0 resection was three times greater with concomitant arterial resections (RR: 3.11; 95% CI [1.65–5.86]; *p* < 0.0227). The studies showed substantial heterogeneity (I^2^ = 68.6%) (Figure 3).

### 3.12. Heterogeneity and Publication Bias

We assessed the studies for potential publication bias through visual examination using a funnel plot. In our result, the funnel plot exhibited asymmetry, indicating the presence of potential publication bias across studies (Figure 4).

## 4. Discussion

This systematic review aimed to provide an update of the current literature on surgery incorporating arterial resections during pancreatic cancer surgery. Our analysis showed that mortality but not morbidity was increased in pancreatic cancer surgery with concomitant arterial resections when compared to standard non-arterial resections. This is different from that reported by Małczak et al. [6], who reported increased overall morbidity with concomitant arterial resections (48.3% vs. 33.7% 95% CI 1.07–1.83; *p* = 0.01). It is important to note, that morbidity in this analysis was driven by higher post-operative bleeding in the concomitant arterial resection group, and that there was no significant difference in the rate of biliary fistulas, post-operative pancreatic fistulas, cardiopulmonary complications and non-R0 rates when compared to standard non-arterial resections. This analysis confirmed no significant increase in serious complications (Clavien–Dindo ≥ 3) with arterial resections with major complications occurring in 31.8% of cases post-arterial resection and 34.3% post-standard resections. This is similar to a recent large single-arm study by Kwon et al. [18], which confirmed the safety of arterial resections with comparable post-operative serious morbidity of 26.6% and mortality of 0.9%.

Mortality associated with arterial resections remains higher than that of standard non-arterial resections. Mollberg et al. [19] in 2001 reported an associated five-fold increased risk of mortality with concomitant arterial resections. The meta-analysis by Małczak et al. [6] included published studies up to 2018 and reported a four-fold increase in mortality. This analysis of recent studies from 2018 to 2024 showed that arterial resections were associated with a three-fold increase in mortality over standard resections (RR: 3.28; 95% CI [0.75–14.46]; *p* = 0.0365). Małczak et al. [6], in 2020, suggested a possible temporal-related decline in both morbidity and mortality in pancreatic surgery with arterial resections. This analysis confirms this notion with a decrease in both morbidity and mortality associated with arterial resections over the past two decades.

Survival data comparing pancreatic cancer surgery with arterial resections compared to non-arterial standard resections are limited and need to be interpreted with caution due to moderate to high heterogeneity found in the reported studies [6].

In this analysis, only three studies [11,13,15] reported on comparative 1-, 2- and 3-year survival rates and only one study Li et al. [15] reported on median overall survival for both groups. There were no significant differences reported in one-year survival rates from any of the three studies included which is similar to the meta-analysis by Małczak et al. [6] where they also found no significant differences in one-year survival rates between the two groups. The three-year survival rate was improved by 40% in the standard non-arterial resection group compared to the arterial resection group in the study by Małczak et al. [6]. In contrast, the 2-year and 3-year survival outcomes in this analysis, while not significantly different, favored the concomitant arterial resection group. The mean 2-year and 3-year survival rates across the three studies included in this analysis were 62.2% and 50.1%, respectively, for the arterial resection group, compared to 55.7% and 49.8%, respectively, for the non-arterial resection group.

These data need to be interpreted with caution. It is important to note that concomitant arterial resections do not equate to histological arterial involvement which would indicate borderline or locally advanced disease. Actual microscopic invasion of resected arteries in the study by Yang et al. [11] was only present in 64.3% of cases and in the study by Li et al. [15] in 61.9%. In addition, the R0 rate reported by Malinka et al. [13] of 40% and 55% is lower than that of other studies [18,20]. This would indicate that there is a proportion of cases included in the concomitant arterial resection group that has tumor staging similar to the standard resection group and are not T4 tumors which carry a possible worse prognosis. In addition, the effect of NAT was not standardized. Patients who underwent concomitant arterial resections had a 2-fold higher chance of receiving NAT which can both increase R0 resection rates and improve oncological prognosis. Further robust studies are needed to confirm results from this analysis on overall survival before conclusive decisions can be drawn.

Pancreaticoduodenectomy with arterial resection for pancreatic cancer involving the head of the pancreas carries a median overall survival rate of 14.8–18.4 months [18,21,22,23]. Hwang et al. [21] reported a significant reduction in median overall survival in pancreaticoduodenectomies with concomitant arterial resections when compared to non-arterial standard pancreaticoduodenectomies (14.8 months vs. 18 months *p* = 0.033). This finding can be explained by the idea that pancreatic cancer involving arteries may represent a more advanced disease compared to localized disease. There are, however, numerous studies and a Cochrane review supporting the survival benefit of concomitant arterial resections with curative intent over palliative treatment [24,25]. This benefit persisted at the 5-year follow-up [25].

Post-operative pancreatic fistula remains one of the major drivers of post-pancreatectomy morbidity and mortality [26]. This analysis confirmed previous studies which found no significant differences in POPF rates in pancreatic cancer surgery with concomitant arterial resections [4,6,27].

Intraoperative blood loss has previously been shown to be an independent risk factor for both overall survival and recurrence-free survival in pancreatic cancer [28]. This analysis confirms that arterial resection and reconstruction can be safely carried out with no increased risk of intraoperative blood loss. Post-operative hemorrhage, although rare, is associated with a significant increase in morbidity and mortality [29]. This analysis showed no significant increase in post-operative hemorrhage with concomitant arterial resections, which could partly be explained by modern advances in hemostasis, surgical technique and intraoperative blood conservation strategies.

Mean R0 resection rates in this analysis were similar to those reported in other studies [6,18]. Similar to studies of concomitant venous resections improving R0 resections [5], in this analysis, the pooled analyses of the four studies [11,13,14,17] included showed arterial resections increased the chance of an R0 resection threefold. As shown by Rompen et al. [20], the R0 resection status is an important independent prognostic variable and could account for the favorable 2- and 3-year overall survival outcomes seen in this analysis.

This analysis was not able to account for the effect of neoadjuvant treatment, the completion rate or the effect of adjuvant treatment. Non-standardized reporting of outcomes post-pancreatic cancer surgery and surgical technique makes conclusive interpretation of outcomes challenging. Future studies controlling for the extent of arterial resection, the histological staging of the malignancy, the oncological biology of the tumor and the effects of NAT and adjuvant treatment need to be addressed in order to make conclusive outcomes.

## 5. Conclusions

Concomitant arterial resections in pancreatic cancer are associated with an increased risk of mortality but not overall morbidity or serious complications when compared to standard non-arterial resections. Recent studies indicated comparative intraoperative blood loss and post-operative pancreatectomy hemorrhage rates between the two groups. Concomitant arterial resections in pancreatic cancer can improve R0 resection three-fold compared to standard resections. In addition, there is a gradual decline in both morbidity and mortality post-arterial resections during pancreatic surgery reported in the literature over time, with studies from 2018 to 2024 indicating acceptable 1- and 3-year survival rates post-arterial resections in selected patients with pancreatic cancer. The observed trend may be influenced by the duration of patient enrolment in some included studies in the meta-analysis. Interpretation of survival data, however, remains cautious due to high heterogeneity amongst studies and limited reported outcome data in some studies. Despite a higher risk of mortality compared to standard resections, the comparative morbidity rates together with acceptable overall survival rates, which are higher than non-operative palliative treatment, should prompt the continued treatment option in high-volume centers in selected patients.

## Figures and Tables

**Figure 1 cancers-17-01540-f001:**
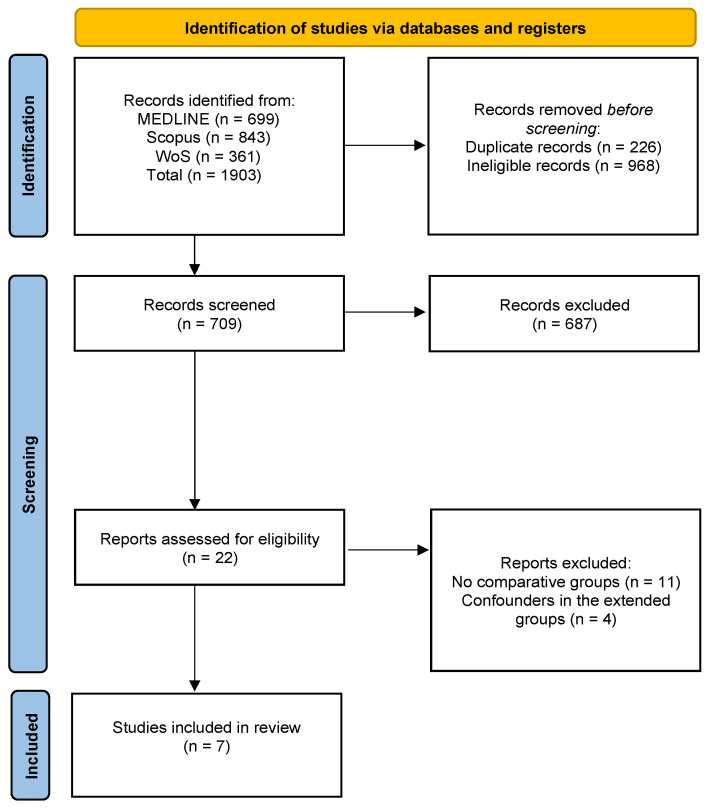
PRISMA flow diagram showing the study selection for the meta-analysis.

**Figure 2 cancers-17-01540-f002:**
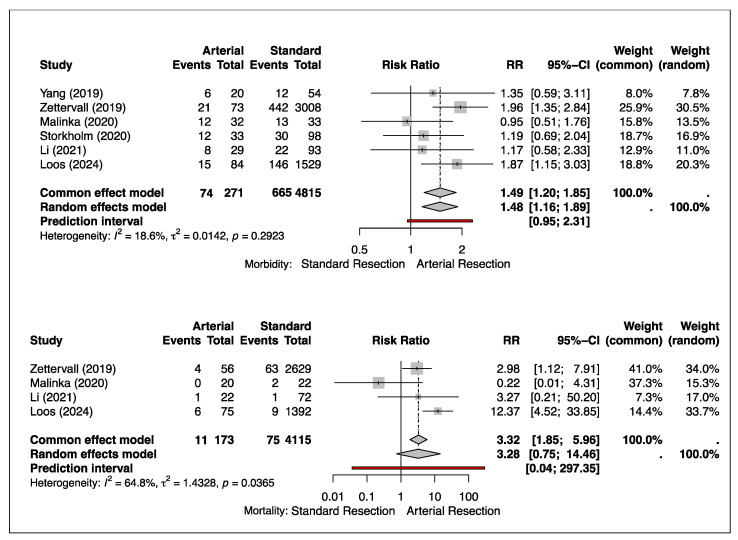
Forest plot shows the relative risk ratio of morbidity and mortality in arterial versus standard non-arterial resections in pancreatic cancer [11,12,13,14,15,16].

**Figure 3 cancers-17-01540-f003:**
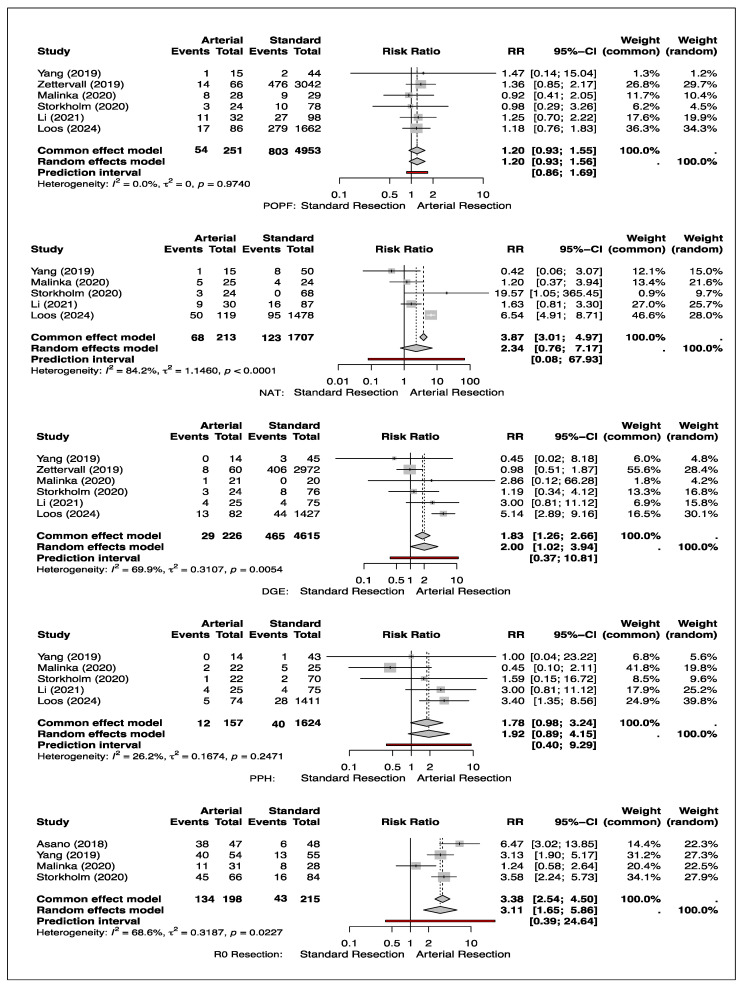
Forest plot showing the relative risk ratio of POPF, NAT, DGE, PPH, and R0 resection in arterial versus non-arterial resections [11,12,13,14,15,16,17].

**Figure 4 cancers-17-01540-f004:**
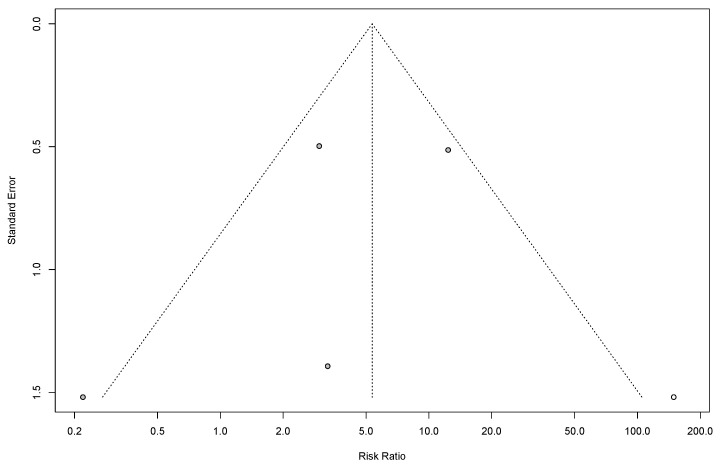
Funnel plot showing the publication bias: prevalence of mortality.

**Table 2 cancers-17-01540-t002:** Median survival and 1-, 2-, and 3-year survival after concomitant arterial versus non-arterial standard resections for pancreatic cancer.

Author	Surgery Type								
	Concomitant Arterial Resection				Non-Arterial Standard Resections				*p*-Value
	Median OS (Months)	1-Year OS (%)	2-Year OS (%)	3-Year OS (%)	Median OS (Months)	1-Year OS (%)	2-Year OS (%)	3-Year OS (%)	
Yang et al.	NS	81.8	63.6	42.4	NS	74	40	38	<0.001
Li et al.	27.4	85	45	30	32.6	94.3	67.1	51.4	<0.001
Malinka et al.	NS	88	78	78	NS	90	60	60	<0.001
Mean (1-, 2-, 3-year OS)		84.9	62.2	50.1		86.1	55.7	49.8	

NS, Not specified; OS, overall survival.

**Table 3 cancers-17-01540-t003:** Comparison of clinical measures with 95% confidence intervals for mean differences.

Variables	*p*	Mean Difference	SE Difference	Lower	Upper
**1-year survival**	0.836	1.167	4.954	−22.482	20.148
**2-year survival**	0.696	6.500	14.391	−55.420	68.420
**3-year survival**	0.980	0.333	11.554	−49.380	50.047
**LOS**	0.874	0.383	2.289	−0.736	0.866
**Blood loss (mL)**	0.709	32.683	79.564	−285.890	220.525
**CDC ≥ 3**	0.738	2.600	7.360	−21.520	16.320

CDC, Clavien–Dindo classification; LOS length of hospital stay; SE, standard error; mL, milliliters.

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
