# Peer review of "Arterial Resections in Pancreatic Cancer—An Updated Systematic Review and Meta-Analysis"

_cancers, 2025, doi:10.3390/cancers17091540_

Round 1
Reviewer 1 Report
Comments and Suggestions for Authors
This is a systematic review and meta-analysis of arterial resection for surgical resection of pancreatic cancer.
- The authors emphasized this analysis was an update one with recent studies. However, despite the inclusion of recent studies, the patient enrollment was longer than 20 years in one large study. Thus, it might be difficult to conclude "there is a gradual decline in both morbidity 431 and mortality post arterial resections during pancreatic surgery reported in the literature 432 over time." The trend should be evaluated by years of surgical resection, rather than publish years.
- There are discrepancies on how to reach consensus in case of disagreement (the entire team vs. a third reviewer; Section 2.2 and 2.3). Please clarify.
- Since this meta-analysis included various studies, defitions of study outcomes should be clarified, such as mortality.
- Section 3.7. Receiving NAT was not a risk factor for arterial resection since NAT would be more administered in cases with arterial invasion. This just means association of NAT and arterial resection.
- Subgroup analyses by surgical resection (PD vs. DP) or NAT (Yes vs. No) might be interesting to readers if possible.
- Spell out all abbreviations in the abstract.
Reviewer 2 Report
Comments and Suggestions for Authors
Dear Authors,
the paper should be improved in this way:
- line 145: you state that 968 papers were not eligible "for various reasons"
- The results presented in Table 2 has no correlation with what you state in the text. Please explain this in a cleares way.
- Line 241. it should be: The pooled data showed "no" significant differences in lenght...
- In the papers showed in table 3, the mean differences in 1-2-3 year of OS between resected and not resected are not so relevant. Please explain why arteria resection should be performed.
- Line 280-281: the sentence Zettervall and Loss... is copied and pasted three times. Please change this
- In the conclusions you state that, according to this reviuew, patients undergoing arterial resections have higher mortality, lower R0 but same OS. What is the advantage to perform resections?
Minor revisions needed.
Round 2
Reviewer 2 Report
Comments and Suggestions for Authors
Dear Authors,
some of the modifications suggested have been performed, other not.
Comments on the Quality of English LanguageMinor revisions needed.